# Towards Semantic Photogrammetry: Generating Semantically Rich Point Clouds from Architectural Close-Range Photogrammetry

**DOI:** 10.3390/s22030966

**Published:** 2022-01-26

**Authors:** Arnadi Murtiyoso, Eugenio Pellis, Pierre Grussenmeyer, Tania Landes, Andrea Masiero

**Affiliations:** 1Université de Strasbourg, INSA Strasbourg, CNRS, ICube Laboratory UMR 7357, 67084 Strasbourg, France; eugenio.pellis@insa-strasbourg.fr (E.P.); pierre.grussenmeyer@insa-strasbourg.fr (P.G.); tania.landes@insa-strasbourg.fr (T.L.); 2Department of Civil and Environmental Engineering, University of Florence, 50121 Florence, Italy; andrea.masiero@unifi.it

**Keywords:** photogrammetry, semantic segmentation, deep learning, automation, dense matching, point cloud, classification

## Abstract

Developments in the field of artificial intelligence have made great strides in the field of automatic semantic segmentation, both in the 2D (image) and 3D spaces. Within the context of 3D recording technology it has also seen application in several areas, most notably in creating semantically rich point clouds which is usually performed manually. In this paper, we propose the introduction of deep learning-based semantic image segmentation into the photogrammetric 3D reconstruction and classification workflow. The main objective is to be able to introduce semantic classification at the beginning of the classical photogrammetric workflow in order to automatically create classified dense point clouds by the end of the said workflow. In this regard, automatic image masking depending on pre-determined classes were performed using a previously trained neural network. The image masks were then employed during dense image matching in order to constraint the process into the respective classes, thus automatically creating semantically classified point clouds as the final output. Results show that the developed method is promising, with automation of the whole process feasible from input (images) to output (labelled point clouds). Quantitative assessment gave good results for specific classes e.g., building facades and windows, with IoU scores of 0.79 and 0.77 respectively.

## 1. Introduction

The use of artificial intelligence has seen an exponential increase in recent decades, aided by developments in computing power. Within the field of 3D surveying, such methods have been used to perform tasks such as semantic segmentation [1]. This process of automatically attributing semantic information into the otherwise geometric information stored in spatial 3D data (e.g., point clouds) is a major step in accelerating the surveying process. Semantic annotation also enables easier modelling and predictions using the available spatial data. Since spatial data annotation is traditionally performed manually, the use of artificial intelligence such as the deep learning approach has the potential to reduce both the significant time and resources required. However, current research mostly focuses on the application of deep learning on the 3D space. In this paper, we propose a method to introduce deep learning semantic segmentation into the classical photogrammetric workflow in order to benefit from some of photogrammetry’s rigorous advantages, e.g., block bundle adjustment.

Photogrammetry as a discipline has a long history of use in the field of surveying. Starting with primarily small scale aerial use [2,3], the use of terrestrial images has also been effectively applied for many applications in larger scale architectural survey [4]. In the past few decades, photogrammetry has also seen significant developments in both its theoretical and technical aspects. Major strides were made in subjects such as analytical photogrammetry [5,6], automatic image matching [7] and bundle adjustment [8]. Furthermore, the parallel development of the computer vision domain such as structure-from-motion [9,10], automatic feature extraction [11] and dense image matching [12,13] have helped in solving some traditional photogrammetric bottlenecks.

Recently, photogrammetry has seen a major democratisation by the advent of low-cost sensors [14,15], more powerful computing capacity [16,17] as well as availability of drones in the public market [18,19]. The latter greatly facilitated close-range photogrammetry as it enabled an aerial point of view which was previously a major constraint in data acquisition. Indeed, 3D reconstruction in general is slowly becoming the standard in mapping, replacing traditional 2D methods and products.

As 3D data both in the form of point clouds and meshes became more and more common as a geospatial product, a new research question rose [20,21,22]. All the major developments in 3D technology, including in photogrammetry, have focused on the geometrical reconstruction of existing objects. This is as far as surveyors are concerned, the main objective of the mapping activity. However, in order to be truly useful, specific tangible meanings must be attached to these geometric elements, i.e., annotating them with relevant semantic information [23,24]. Semantic information or attributes will give these 3D data richness and opens the possibility for various spatial analysis and modelling. Within the traditional 2D mapping, one of the most known framework for this mixture of geometry and semantic attribute is the Geographical Information System (GIS) [25,26]. Its extension into the 3D space can be seen in, for example, 3D GIS for smaller scale scenes [27,28] or Building Information Models (BIM) in larger scenes [29].

The problem of attribute annotation into geometric data was mostly addressed manually [30]. Indeed, historically, GIS layers were physical maps which were digitised and vectorised. When required, semantic annotation was performed at the same time by the operator. This method continued on with the arrival of BIM, where most users would create parametric 3D models from point clouds and attach attributes to them manually [31]. Attempts at automation can be seen in the current literature [32,33,34], and remains a major research question today. In practice, this process of data labelling translates into 3D data classification in geomatics or semantic segmentation in AI parlance [35].

Various methods for semantic segmentation have been proposed in the literature, with some review papers highlighting this fact [1,36]. Techniques based on heuristic information (e.g., geometric rules or tendencies for certain object classes) present a fast and generally precise results [37,38]. These algorithmic approaches are however often non-flexible and problems may occur when encountering complex cases, e.g., historical buildings or traditional architecture. More recent research into artificial intelligence, coupled with more powerful computing power, has opened the possibility to the application of machine learning (ML) to fulfil this purpose. Deep learning (DL), a subset of machine learning, has also seen major strides in performing semantic segmentation on 2D images [39]. Promising results can also be observed in 3D semantic segmentation, both indirectly [40] and directly [41]. It is worth noting that in DL-based solutions, a major bottleneck is the availability of labelled data for training. While over the past few years immense amounts of labelled images have become available in ML circles, 3D training data remain scarce [23] due to the higher complexity in manually labelling them.

While these solutions show promise, most are concerned with the segmentation of the point clouds or 3D meshes which are the product of 3D reconstruction techniques. In this regard, the process of creating these 3D inputs for classification matters a little as they may come from either photogrammetry, lidar, a combination of both, or other 3D sensors. Few studies (e.g., [42]) had addressed the potential of involving this semantic segmentation process directly within the photogrammetric workflow. Hypothetically speaking, such integration may benefit from several advantages. For instance, the application of semantic segmentation on the input images may take advantage from the vastly more available 2D training datasets for DL. Furthermore, when applied on the dense image matching step, other mathematical conditions such as the epipolar constraint and error minimisation via bundle adjustment may help improve the results.

The idea of using AI to support photogrammetry has been previously explored by several studies. In [43], the authors reported several applications including aiding feature detection for the orientation of images with significant differences in scale and viewing angle. A study by [44] presented a similar approach to the one presented in this paper, i.e., using AI to generate classified point clouds from 2D images via photogrammetry although the authors did not prove numerical assessments of their method. The same authors also briefly reported their experiments in using masks to automatically clean dense point clouds, as well as to transfer 2D labels to 3D point cloud [42]. A similar approach using masks was reported in [45] for agricultural applications. Furthermore, the authors in [46] attempted to implement masks during dense matching in order to clean point cloud noise.

The aim of the study is therefore to propose a method which may benefit from both the abundance of 2D training data for DL purposes and the rigour of photogrammetric computations to create a faster and more precise approach to 3D point cloud semantic segmentation. To this end, in this paper we propose a practical and fully automatic workflow from images to classified 3D point clouds. As a proof of concept, the workflow was implemented within the context of close-range photogrammetry for architectural surveying purposes, e.g., building facade modelling. The input of the said workflow is 2D images acquired according to photogrammetric principles and oriented using a classical image matching and bundle adjustment. A DL-based neural network trained on a database of rectified building facade images was then used to perform semantic segmentation on the oriented images. The segmented images were thereafter used in dense image matching to generate semantically rich and classified 3D point cloud. The workflow was implemented using the open source photogrammetric suite Apero-Micmac [47], with additional coding in Matlab. Additional comparison was also implemented in the commercial software Agisoft Metashape. As the readers shall observe in this paper, the proposed method may be adapted into other photogrammetric situations, e.g., aerial mapping or heritage documentation by simply adjusting the applied neural network. As far as the paper’s structure is concerned, the next section will explore some work related to the main idea presented in this paper. Section 3 will thereafter contain the main description of the proposed method, with experimental results and discussions presented in Section 4 and Section 5, respectively. Finally, Section 6 shall put forward arguments to the potential of the proposed method, its drawbacks, and some ideas for improvements.

## 2. Literature Review

In the following exposition, a summary of existing literature on the subject of photogrammetry, AI semantic segmentation and their interaction shall be addressed. First, an overview of the photogrammetric workflow will be described. Arguments will also be put forward on the choice of software solution used in this study. Subsequently, a short description of deep learning methods for 3D semantic segmentation will be given. Finally, several existing solutions to the problem of projecting 2D image labels into the 3D space will be described.

### 2.1. Notions on Photogrammetry

Photogrammetry as a mapping technique attempts to convert 2D images into 3D coordinates using stereo vision principles [48]. While such concepts were first implemented in an empiric manner, as is the case with analogue photogrammetry, mathematical relations were soon developed to enable an analytic approach to the problem of 3D reconstruction. Notably, the collinearity and coplanarity conditions played an important role in establishing a relation between the 2D and 3D space [6]. For most of its history and even today, photogrammetry remains very focused on the problem of precision. This is in line with the original objective of photogrammetry as a remote sensing mapping tool. However, almost in parallel developments in the computer vision domain saw significant leaps as evidenced by the popularity of Structure-from-Motion as a solution to image pose estimation [10]. This progress, in addition to other developments in both imaging sensor and computing technologies, enabled the unprecedented automation of the traditional photogrammetric workflow albeit sometimes at the expense of rigorous quality control [49]. Image matching algorithms further reduce the necessity of manual measurements, e.g., those involving the traditional six Von Grüber points [11,50].

Evolving from previous solutions for the problem of aerotriangulation, i.e., densification of ground controls [50] in analytical photogrammetry, the concept of bundle adjustment refers to the simultaneous computation of image exterior orientation parameters (also referred to as extrinsic parameters in computer vision [10,35]) and point coordinates in the 3D space. It typically involves a non-linear optimisation calculation based on either collinearity or coplanarity equations. This simultaneous “block” adjustment of the whole system provides a rigorous solution for the exterior orientation problem [49]. The bundle adjustment may also include the resolution of camera internal parameters in a process called self or auto-calibration [51]. Furthermore, modern bundle adjustment solutions may include damping techniques (e.g., Levenberg–Marquardt algorithm) to help the classical Gauss–Newton least-squares method in reaching final convergence [8]. This is the case, for example, in the software Apero-Micmac used in this study [47].

Another major breakthrough in the field of photogrammetry was the development of dense image matching. Work on Patch-based Multi View Stereo (PMVS) [9] and Semi-Global Matching [52] may be considered some of the most important developments. Dense image matching is a crucial development for photogrammetry which enables it to generate dense point clouds not unlike those created by lidar. This provides photogrammetry with the tool to compete with lidar systems [53], although in practice they are often complementary, especially in large-scale applications [54].

Various photogrammetric solutions exist in the market today, both of commercial and open source nature. A classical photogrammetric workflow starts with the acquisition of images. Certain rules must be respected in order to guarantee good results from photogrammetry, e.g., enough overlapping between images [55], configuration of image network [49,51,56] but also photographic quality [57]. From a surveying perspective, pre-acquisition steps such as determination of the required Ground Sampling Distance (GSD) [21] and distribution of Ground Control Points (GCP) or scale lines are equally important [18,58]. Image orientation with bundle adjustment is then usually performed before continuing with dense image matching in order to create dense point clouds.

However, these point clouds for the most part represent only the geometric aspect of the object in question. Semantic information is usually imbued by performing point cloud classification [59,60] as a post-processing of the point cloud generation process. Indeed, most studies including those with application of DL involve semantic segmentation on the point cloud [1]. In this paper, DL-based methods are introduced during the photogrammetric process with the final goal of creating a truly semantic photogrammetry workflow.

### 2.2. DEEP Learning for Semantic Segmentation

Semantic segmentation refers to the process of grouping parts of a data into several subsets that share similar feature characteristics. It can be considered as a fundamental step in the machine automatic comprehension and it is a key topic in a lot of computer vision problems such as scene understanding, autonomous driving, remote sensing, robotic perception, and many others. The continuously increasing number of applications concerning semantic segmentation makes it a very active research field, and different methods and approaches are proposed every year. Image segmentation, or 2D semantic segmentation, involves a pixel-level classification, in which each pixel is associated with a category or a class. Point cloud semantic segmentation is the extension of this task in the 3D space, in which irregular distributed points are used instead of regular distributed pixels in a 2D image. Point cloud semantic segmentation is usually realised by supervised and unsupervised learning methods, including regular learning and deep learning [61]. In the last five years DL on point clouds has been attracting extensive attention, due to the remarkable results obtained on two-dimensional image processing, in particular after the introduction of Convolutional Neural Networks (CNN). Compared with two-dimensional data, working with 3D point clouds provides an opportunity for a better understanding of spatial and geometrical information, and a better comprehension and characterisation of complex scenarios. However, the use of deep learning on 3D point clouds still faces several significant challenges due to: (i) the unstructured and unordered nature of point clouds, which prevents the use of 2D network architectures, (ii) the large data size, which implies long computing time and (iii) the unavailability of large dedicated dataset for the networks training process. Studies exist which aim to remedy this problem [62].

Despite these challenges, more and more methods are proposed to work with point clouds. In the current literature, semantic segmentation techniques for 3D point cloud can be divided into two groups: (i) projection-based methods and (ii) point-based methods [63].

#### 2.2.1. Projection-Based Methods

The main issue to solve in the problem of point cloud segmentation using standard neural network is its unstructured nature. To address this issue, projection-based techniques first apply a transformation to convert 3D point cloud into data with a regular structure, before subsequently performing semantic segmentation by exploiting the standards models and finally re-projecting the extracted features back to the initial point cloud. Although intermediate representation involves inevitably a spatial and geometrical information loss, the advantage of these methods is the ability to leverage well-established 2D network architectures. According to the type of representation, it is possible to distinguish four categories among these methods:Multiview representation: These methods project firstly the 3D shape or point cloud into multiple images or views, then apply existing models to extract feature from the 2D data. The results obtained on the image representation are compared and analysed, and then re-projected on the 3D shape to obtain the segmentation of the 3D scene. Two of the most popular works are MVCNN [64] which proposed the use of Convolutional Neural Networks (CNN) on multiple perspectives and SnapNet [65] which uses snapshots of the cloud to generate RGB and depth images to address the problem of information loss. These methods ensure excellent image segmentation performance, but the 3D features transposition remains a challenging task, producing large loss of spatial information.Volumetric representation: Volumetric representation consists in the transformation of the unstructured 3D cloud into a regular spatial grid, a process also called voxelisation. The information as distributed on the regular grid is then exploited to train a standard neural network to perform the segmentation task. The most popular architectures are VoxNet [66] which uses CNN to predict classes directly on the spatial grid, OctNET [67] and SEGCloud [68] which introduced the methods of spatial partition such as K-d tree or Octree. These methods require large amounts of computing memory and produce reasonable performance on small point clouds. They are therefore unfortunately still unsuitable for complex scenarios.Spherical representation: This type of representation retains more geometrical and spatial information compared to multiview representation. The most important works in this regard include SqueezeNet [69] and RangeNet++ [70] especially for application on real time lidar data segmentation. However, they still have to face several issues such as discretisation errors and occlusion problems.Lattice representation: Lattice representation converts a point cloud into discrete elements such as sparse permutohedral lattices. This representation can control the sparsity of the extracted features and it reduces memory requirement and computational cost compared to simple voxelisation. Some of the main studies include SPLATNet [71], LatticeNet [72] and MinkowskiNet [73].

#### 2.2.2. Point-Based Methods

Point-based methods do not introduce any intermediate representation, and they work directly with point clouds. This direct approach leverage on the full use of the characteristics of the raw cloud data and consider all the geometrical and spatial information. They seem the most promising but are still in development and they still have to face several critical issues. Overall, these methods could be divided into four groups:Pointwise methods: The pioneering work for this method is PointNet [74] which learns per-point features using shared Multi-Layer Perceptrons (MLPs) and global features using symmetrical polling functions. Since MLP cannot capture local geometry, a lot of networks based on PointNet have been developed recently. These methods are generally based on neighbouring feature pooling such as PointNet++ [75].Convolution methods: These methods propose an effective convolution operator directly for point clouds. PointCNN [76] is an example of a network based on parametric continuous convolution layers and kernel function as parameterised by MLPs. Another example is ConvPoint [77] which proposed a point-wise convolution operator and convolution weights determined by the Euclidean distances to kernel points.RNN-based methods: Recurrent Neural Network (RNN) are used recently for the segmentation of point clouds, in particular to capture inherent context features. Based on PointNet, they first transform a block of points into multi-scale blocks or grid blocks. Then the features extracted by PointNet are fed into a Recurrent Consolidation Units (RCU) to obtain the output-level context. One of the most popular networks in this regard is 3DCNN-DQN-RNN [78].Graph-based methods: Graph Neural Network (GNN) is a type of network which directly operates on graph structure. Several methods leverage on graphs to capture richer geometrical information, for example DGCNN [79].

Point-based methods seem to be the most promising in the future as evidenced, amongst others, by the great interest it generated in recent research. However, this study will focus more on the deployment of a workflow for dense point cloud semantic segmentation based on two-dimensional data as integrated within the traditional photogrammetric workflow. On one hand, this approach allows us to exploit the tried-and-tested results in 2D image processing while on the other hand it allows the automatic creation of a directly segmented and classified point cloud. In addition, the interaction between point clouds and images could converge in a hybrid point-image method that may improve the performance of both approaches in the future.

### 2.3. Reprojection of 2D Semantic Segmentation into the 3D Space

In the case of multiview deep learning approaches for 3D semantic segmentation, two main steps may be distinguished: (i) the labelling of the two-dimensional images related to the 3D scene, and (ii) the (re)projection of such labels from the images to the 3D shape or point cloud. Since numerous techniques and methods are already developed for 2D image segmentation with promising results and accuracy [80], the most challenging and critical step in this framework is the reprojection step. This operation introduces inevitably a loss of spatial and geometrical information, and, in many cases, involves a loss of accuracy on the overall performance.

In the last years, several methods have been proposed to address these problems. In [81], the authors proposed a 2D-to-3D based label propagation approach to create 3D training data by utilising existing datasets such as ImageNet and LabelMe. The proposed method consists of two major novel components, Exemplar SVM based label propagation, which effectively addresses the cross-domain issue, and a graphical model based contextual refinement incorporating 3D constraints.

For similar purposes the method developed in [82] propagates object label from 2D image to a sparse point cloud by matching a group of points that corresponds to the area within the 2D bounding box in the image. Furthermore, [42] proposed a semantic photogrammetry workflow similar to the one proposed in this paper, in which the label back-projection is based on the projection matrix which connects the 3D with the 2D space. Using this approach, all of the images contribute to the labelling projection on the cloud with a weighted winner procedure. Although the proposed method is similar, the authors only described their method briefly with few quantitative analysis.

Our previous work described in [40] presented an approach for the segmentation of 3D building facade based on orthophoto and the corresponding depth maps. The XY coordinates of each pixel in the orthophoto was used to determine the corresponding planimetric coordinates of the point in the point cloud and finally a winner-takes-all approach was applied to annotate the 3D points with the respective 2D pixel class.

In [83], the authors proposed an approach for label propagation in RGB-D video sequences, in which each unlabelled frame is segmented using an intermediate 3D point cloud representation obtained from the camera pose and depth information of two keyframes. For similar purposes some studies deal with the 3D to 2D projection as can be seen for example in [84]. In this paper, a CFR model was proposed which is able to transfer the labels from a sparse 3D point cloud to the image pixels by leveraging the calibration and the registration of a camera and laser scanner system, estimated using structure-from-motion. Finally, the authors in [85] developed a method to map the semantic label of 3D point clouds into street view images. The images are over-segmented into super-pixels, and each image plane super-pixel is associated with a collection of labelled 3D points using the generic camera model.

## 3. Proposed Method

Figure 1 presents a flowchart of the developed workflow. It starts with image acquisition following standard photogrammetric procedure. The acquired images were then processed using the previously trained DL network to semantically segment them according to the predetermined classes. The output of this process is class labels for each pixel for each input image. Using these segmented images, class masks were then generated which was later on used as constraints during the dense image matching process. The final result would be a semantically segmented 3D dense point clouds directly out from the photogrammetric process without need for further labelling or annotation.

In the case of this paper, image acquisition of a building facade was conducted using terrestrial images as a proof of concept for the semantic photogrammetry method. The building used in this case is the main facade of the Zoological Museum of Strasbourg, France. The dimensions of this facade is roughly 40 × 10 m. Note that the building was built in the 19th century and therefore presents a typical architecture of the era; indeed, it is part of the UNESCO World Heritage site of Neustadt since 2017. This heritage aspect is another challenge for the DL networks, since heritage architectural elements are more complex and thus more difficult to identify [23]. In this case, the terrestrial images were taken using a Canon EOS 6D DSLR camera with a 24 mm fixed lens.

A total of 33 images were acquired and processed using the open source Apero-Micmac software suite [47]. As an additional comparison, they were also independently processed using the commercial software Agisoft Metashape. The use of Apero-Micmac in this study is prioritised since almost if not all theoretical aspects of this open source suite can be determined and more importantly verified, whereas the same cannot be said of commercial solutions for understandable reasons related to trade secrets. That being said, Metashape also employs a bundle adjustment computation process [49] and an SGM-like dense matching approach [53].

Parallel to the computation of the image orientation parameters, a neural network was applied on the input images to semantically segment them. To this end, a DL network of the DeepLabV3+ architecture [86] pre-trained using a ResNet-18 network [87]. Using the pre-trained network, further training was performed using an open dataset prepared by the Center for Machine Perception (CMP) of the Czech Technical University [88]. This dataset consists of 606 rectified images of building facades with varying types of architecture. This process of transfer learning was deemed adequate to perform the 2D semantic segmentation of the case study presented in this paper. Furthermore, the images were classified into six classes: “pillar”, “door”, “facade”, “window”, “shops” and “background”. The “shops” class refers to business signs and plaques. Note that this setup is more or less identical to a previous study as referred in [40].

Once the semantic segmentation was performed on all the input images, a simple script enabled the extraction of pixels pertaining to each class. Image masks were created for each class and for each image (Figure 2). These masks were then integrated into the photogrammetric process by applying them during the dense matching step. At this stage, it is assumed that the exterior orientation parameters acquired from the bundle adjustment are of a good quality. Dense matching was thereafter performed separately for each of the six classes using the image masks as constraints. The result is six distinct 3D dense point clouds which will naturally inherit the classes of the respective input 2D masks.

## 4. Experimental Results and Assessments

In the following section the case study on the Strasbourg Zoological Museum will be presented. A visual description of some of the results can be seen in Figure 3, in which dense point cloud generated by Micmac is presented, along with the manually segmented ground truth and the prediction results. The outcome of the same method applied in Metashape is also presented in said figure. It should be noted that the ground truth displayed in Figure 3 is created from manual segmentation of Micmac point cloud. A separate ground truth was also created for the Metashape point cloud.

In order to perform quantitative assessment on the results, several metrics were chosen to measure the performance of the proposed method. The semantic segmentation metrics of precision, recall and the aggregate F1 score were used in this regard. In addition, the Intersection over Union (IoU) score was also used to assess the results. As has been previously mentioned, for each photogrammetric software a separate ground truth was created. These ground truth data were created from combining all the separate point clouds generated by the method as described in Figure 1 and Figure 2, and then manually labelled.

Table 1 shows the confusion matrix for the proposed semantic photogrammetry method applied to the software Micmac. Note that the assessment does not include the “background” class which was not considered particularly pertinent overall (see, however, a technical application in Section 5.3). In general, the proposed method seems to show promising results judging from the number of correctly classified points. Similarly Table 2 shows the same matrix for Metashape. In addition, Figure 4 presents a comparison between the statistics obtained from both Micmac and Metashape. In both cases, the proposed method was able to perform well in detecting and segmenting important classes such as windows and doors. The good performance on the facade class is nevertheless expected since it constitutes the majority of labels in any building-related semantic segmentation. The results for the “shops” class, in this case defined as panels and business signs, seem to be better in Metashape than Micmac. This point may be further explained by the fact that Metashape generated a much denser point cloud than Micmac. Indeed, this issue might be related to the fact that Micmac employs a much stricter post-filtering of dense matching than Metashape in relation to problematic areas such as little or textureless objects and shadows [18].

The results for the building openings (i.e., windows and doors) are especially encouraging because this has often been known to be a particular problem in building semantic segmentation, especially those using point-based approaches [89]. These results become even more interesting in light of the many potential applications for the automatic detection of building openings, such as automatic indoor–outdoor point cloud registration [90] or BIM creation [31]. For this reason, a further comparison was performed between these results and our implementation of PointNet++, which shall be detailed in Section 5.1. Furthermore, a comparison against another approach developed in a prior work shall also be explained in the next section.

## 5. Discussion

### 5.1. Comparison to Previous Work

In previous work detailed in Murtiyoso et al. (2021) [40], we presented another approach to reprojection-based semantic segmentation. While the neural network was prepared in a similar manner, in this paper the pixel class prediction was performed on the orthophoto of a building facade instead of the input images as presented in this paper. This method produced good results also for the building openings, but was severely limited by the fact that an orthophoto and a depth map are required as inputs. This may prove problematic in the case of more complex building architectures, hence the development of the semantic photogrammetry method as described here. In this section, a comparison is performed between the method proposed in this research and the one described in our previous work.

Furthermore, in another experiment conducted almost in parallel to the development of the methods in this paper, an implementation of the PointNet++ architecture was done for the Zoological Museum dataset [91]. This enables a further comparison to a point-based 3D segmentation method in order to better assess the results of this study. For the PointNet++ implementation, the 3D point cloud of the four facades was acquired. All of the facades were then manually labelled into classes. Three facades were then used to train the neural network, with the fourth and final facade used as a test data. In this case, the same main facade as the one used in this paper and in [40] was used.

For the purposes of this comparison, only three classes (“window”, “door” and “facade”) shall be compared since both the “shops” and “pillar” classes were grossly underrepresented in the training data for PointNet++. This is owed to the fact that within the Zoological Museum dataset these two classes do not present adequate data, whereas they are not negligible in the CMAP image dataset used for the training in this paper.

Figure 5 describes the comparison between these methods in a histogram representation. As can be observed from the figure, the proposed method shows a clear advantage in regards to PointNet++. Indeed, for PointNet++ the “door” class is virtually non-existent while the “window” class IoU score is less than 0.5. As has been mentioned before, this is a known issue in direct point-based 3D segmentation. The main reasons are usually related to inadequacy in terms of training data and point features, especially in the case of building openings. Compared to our previous approach in [40], the proposed semantic photogrammetry method presented an improvement with regards to the two building opening classes while the detection of facade remained better in this previous approach. However, this previous method is very limited to certain buildings with mostly flat facades and few architectural ornaments. The need to acquire not only the orthophoto but also a depth map to reproject the labels to the 3D point cloud may also present additional problems. This would be the case especially in heritage buildings with more complex types of architecture.

### 5.2. Comparison to Other Studies

Finally, in order to further assess the results obtained especially in the case of building openings, a comparison was also performed to other studies which use AI-based semantic segmentation to perform the detection of openings, i.e., windows and/or doors. Four papers were identified, all of which were based on either an ML or DL, and are fairly recent. In Malinverni et al. (2019) [89], the authors used DGCNN to perform the task. Building upon this, Pierdicca et al. (2020) modified the base DGCNN architecture [41]. Matrone et al. (2020) [34] presented results not only from this modified DGCNN, but also the inclusion of 3D features during training. Finally, Grilli et al. (2020) [92] presented some results from their implementation of Random Forest (RF) algorithm.

Throughout these studies, the “window” and “facade” class was the only one common to all of them. Figure 6 shows therefore the comparison on the performance of each method in a histogram form. For comparison purposes, values for Grilli et al. (2020) and Matrone et al. (2020) represent the average of the several datasets described in those papers. Furthermore, results of the modified DGCNN with 3D features in Matrone et al. (2020) was chosen for this comparison. Similarly, the values for the proposed method present an average of results from both Micmac and Metashape. It should be noted that this comparison is only intended as a general overview, since for each study not only the method is different but also the nature of the case studies, the training data and their distribution of class labels as well as the determination of which classes were included during the segmentation. From Figure 6, the semantic photogrammetry method proposed here seems to have an advantage at least for the “window” class.

Overall, the proposed method registered better scores compared to the four other studies using AI for the semantic segmentation of building openings. It is worth noting that the four studies included in the comparison are all based on point-based semantic segmentation, i.e., direct segmentation of the 3D point cloud. In the majority of these cases, classes representing building openings e.g., windows are often underrepresented, as can be seen in our own implementation of PointNet++ described previously in Section 5.1. On the other hand, facade or walls are mostly overrepresented, although in some of the cited studies the authors further divide the facade into several other classes, e.g., mouldings and vaults. This is reflected by the results from the three DL-based approaches of Malinverni et al. (2019), Matrone et al. (2020) and Pierdicca et al. (2020), as shown in Figure 6. However, using more classical Random Forest ML-based approach, Grilli et al. (2020) were able to achieve better results in the case of windows. The implemented semantic photogrammetry approach was able to outperform all other studies for the detection of building openings, while reaching a comparable result to RF in the case of facades.

Furthermore, it may be argued that in these other cases, the source of the point cloud is irrelevant due to the point-wise nature of the segmentation. Using the proposed approach, we argue that both the much more available training data for 2D segmentation and the introduction of the DL process into the photogrammetric workflow directly contribute to the observed performance. It is also interesting to note that the current implementation of semantic photogrammetry as described in this paper involves a small training dataset for DL standards (606 images), and further improvements and adaptations of this proof of concept may be envisaged in the future.

### 5.3. Example of Direct Application: Point Cloud Cleaning

In order to show the potential of the developed approach, an example of direct application can be seen in Figure 7. In this figure, the semantic photogrammetry approach was used to automatically mask unwanted objects in a scene, directly from the 2D images input. Concretely, this involves the inversion of the masks for the “background” class, thus excluding objects not considered as of interest. Furthermore, this approach for automatic point cloud cleaning not only excludes unwanted object classes, but may also reduce overall processing time during dense image matching. This is because the masks by virtue of its constraining effect reduces the area of interest to be matched. Quantitative assessment has shown that this method manages to achieve a 0.86 F1 score for the non-background classes (all combined).

## 6. Conclusions and Future Investigations

This paper presents an approach to introduce AI-based semantic segmentation into the photogrammetric workflow, in an attempt to develop a semantic photogrammetry method. The proposed method takes benefit from the abundance of 2D image label data and reliable AI-based methods available today, in contrast to the scant availability of 3D labelled point clouds especially for large scale applications. With semantic segmentation performed on the 2D input images, a processing strategy based on the creation of 2D image masks were developed. The image masks created correspond to the class labels, and create therefore separate point clouds for each class.

The proposed method was implemented in both Apero-Micmac and Metashape. While the comparison of these two pieces of software in their capacity as photogrammetric solutions is beyond the scope of the paper, it has been shown that the quality of dense matching also plays a role in the final quality of the result. Furthermore, the post filtering process also plays a role as it determines the level of noise, i.e., false positives in the final dense point cloud. This relation between semantic photogrammetry and dense image matching quality has not been sufficiently investigated and may be an interesting subject for a future work.

Nevertheless, this paper attempted to present a proof of concept to the possibility to use AI in photogrammetric task. In the case study and comparisons, this was demonstrated in the case of building facade segmentation. The method has shown that the initial hypothesis of using the vastly more available labelled 2D training data is beneficial, as highlighted in the comparisons. Especially for the very interesting application of building opening detection, the proposed method has performed well. On the other side, this has also shown the limitation of the current implementation of the approach. Indeed, underrepresented classes, e.g., shop signs and pillars still pose problems although this is a more general problem with any method of semantic segmentation.

Based on the results obtained in the experiments, the developed method of semantic photogrammetry show much promise. It is also interesting to investigate its potentials for implementations in other settings, e.g., aerial photogrammetry, building interior modeling or even low-cost spherical photogrammetry. Evidently different scenes will require different sorts of DL learning; however, the overall semantic photogrammetry approach may be easily transposed on these different scenes thereafter.

Other points for improvement include the generation of 2D training data more suited to the encountered situation. For example, in this study the CMP database was used to train the neural network. This image database consists of rectified images, i.e., images already processed to have a perpendicular point of view. This does not exactly correspond to the input images in the experiments, which were close-range photogrammetry images. Methods to automatically create more suitable training data for close-range photogrammetry are also under investigation, with preliminary results described in [62].

## Figures and Tables

**Figure 1 sensors-22-00966-f001:**
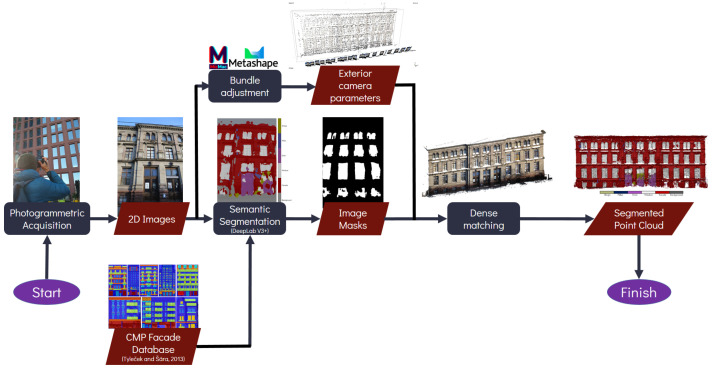
Developed workflow for the proposed semantic photogrammetry process.

**Figure 2 sensors-22-00966-f002:**
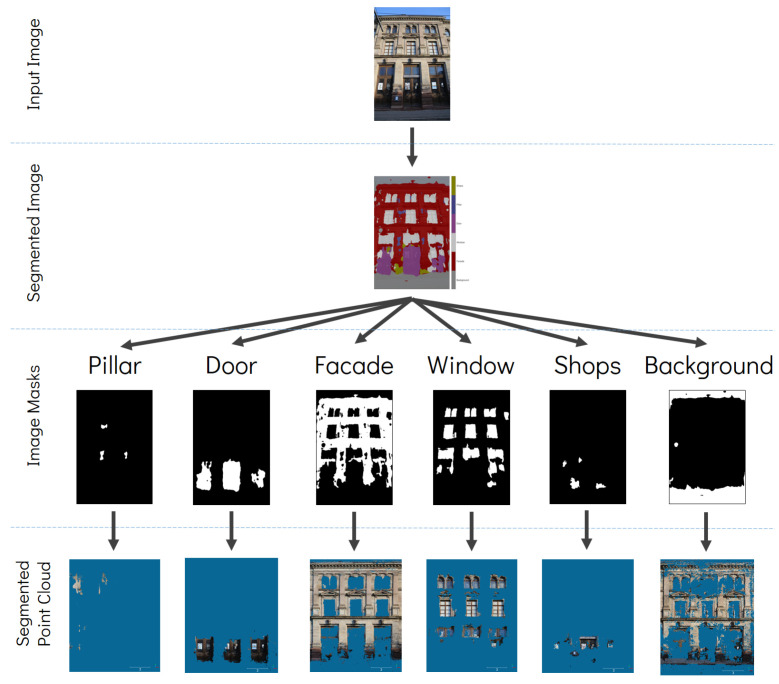
Creation of class-dependent image masks from the segmented image and its application in dense matching to generate semantically classified point clouds.

**Figure 3 sensors-22-00966-f003:**
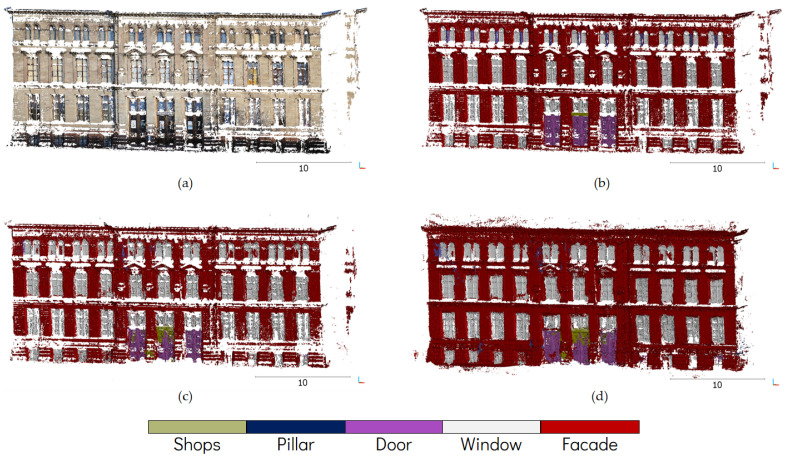
Visual illustration of some results from the experiment: (**a**) raw unclassified dense point cloud generated by Micmac, (**b**) manually segmented ground truth, (**c**) result of the semantic segmentation on Micmac dense point cloud and (**d**) result of the same procedure applied to Metashape dense point cloud.

**Figure 4 sensors-22-00966-f004:**
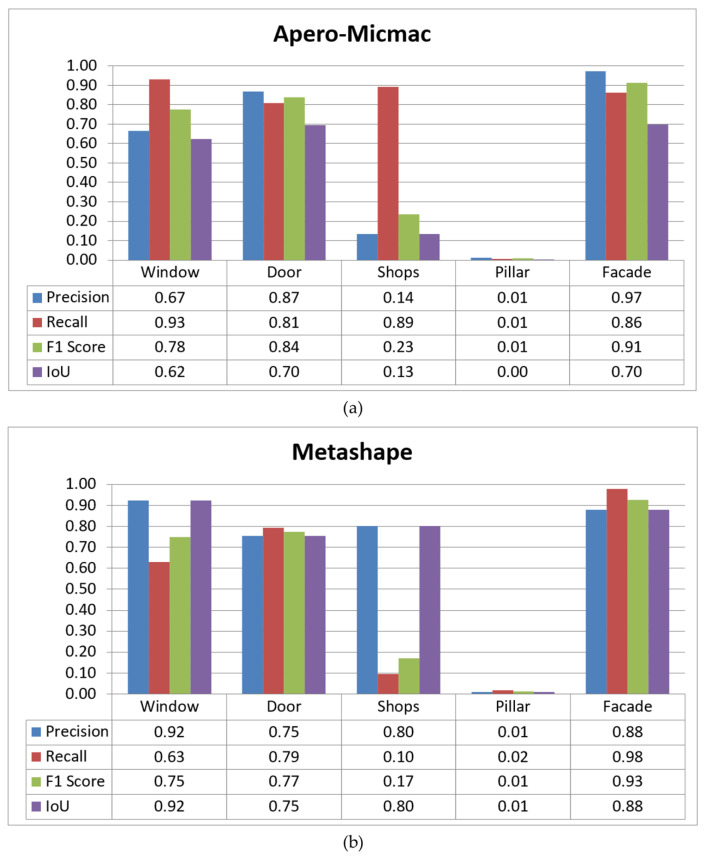
Performance statistics for the proposed method applied to dense point clouds generated by (**a**) Micmac and (**b**) Metashape.

**Figure 5 sensors-22-00966-f005:**
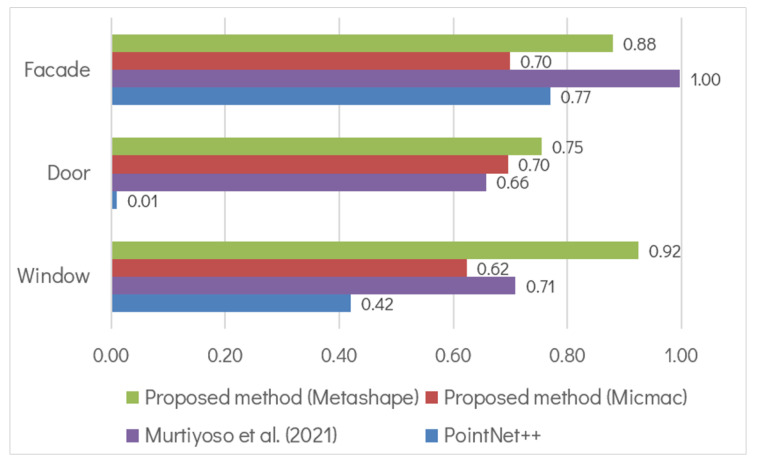
Comparison of IoU scores of the proposed method to other previous work.

**Figure 6 sensors-22-00966-f006:**
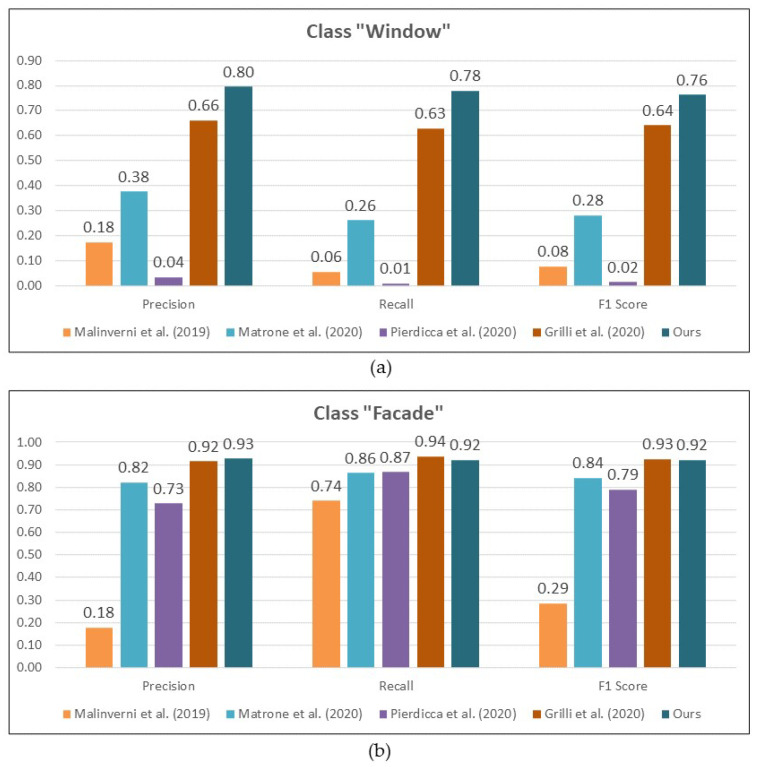
Comparison to other studies for the class (**a**) “window” and (**b**) “facade”.

**Figure 7 sensors-22-00966-f007:**
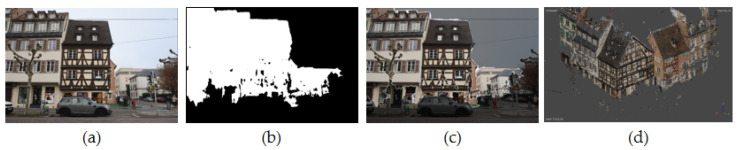
Example of concrete application of the proposed method in photogrammetric point cloud cleaning: (**a**) original image, (**b**) mask of all classes except “background”, (**c**) mask applied to the original image and (**d**) 3D point cloud from dense image matching using the masked image.

**Table 1 sensors-22-00966-t001:** Confusion matrix for the semantic segmentation on Micmac dense point cloud.

		**Ground Truth**					
	* **Class** *	**Window**	**Door**	**Shops**	**Pillar**	**Facade**	* **Total** *
**Predicted**	**Window**	**548,886**	3920	688	14,927	254,941	*823,362*
	**Door**	1566	**152,538**	0	0	21,483	*175,587*
	**Shops**	682	25,738	**6216**	0	13,326	*45,962*
	**Pillar**	15	0	0	**124**	9535	*9674*
	**Facade**	38,903	6121	66	6910	**1,876,172**	*1,928,172*
	* **Total** *	*590,052*	*188,317*	*6970*	*21,961*	*2,175,457*	* **2,982,757** *

**Table 2 sensors-22-00966-t002:** Confusion matrix for the semantic segmentation on Metashape dense point cloud.

		**Ground Truth**					
	* **Class** *	**Window**	**Door**	**Shops**	**Pillar**	**Facade**	* **Total** *
**Predicted**	**Window**	**3,104,942**	22,286	6467	6697	217,949	*3,358,341*
	**Door**	42,294	**819,405**	179,411	87	44,619	*1,085,816*
	**Shops**	6427	0	**28,283**	0	531	*35,241*
	**Pillar**	181,417	0	0	**2621**	54,961	*238,999*
	**Facade**	1,595,515	190,429	82,260	130,134	**14,612,534**	*16,610,872*
	* **Total** *	*4,930,595*	*1,032,120*	*296,421*	*139,539*	*14,930,594*	* **21,329,269** *

## Data Availability

Not applicable.

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
