# Peer review of "Towards Semantic Photogrammetry: Generating Semantically Rich Point Clouds from Architectural Close-Range Photogrammetry"

_sensors, 2022, doi:10.3390/s22030966_

Round 1

Reviewer 1 Report

Reviewer Comments

Paper title: Towards semantic photogrammetry: generating semantically rich point clouds from architectural close range photogrammetry

The present manuscript describes an integration of deep learning-based semantic image segmentation with photogrammetric 3D reconstruction and classification.

A manuscript has a practical application and also provide important theoretical for the next studies.

The paper can be accepted for publication after providing the corrections mentioned below.

Point 1. In the Introduction It will be great if the authors show some description in context – Why it is important to conduct this study?

Point 2. The aim and the tasks must be highlighted at the end of the Introduction section.

Point 3. 'The state-of-the-art' is not presenting up-to-now research which is the main topic of this contribution. I would rather remove this sub-heading and keep only the 'Introduction' chapter or combine it with the text that follows.

Point 4. It is strongly recommended to follow the IMRaD structure. Now it is difficult to understand the Methods of research. It is unclear.

Point 5. Figure 4 and Figure 5 seem to be tabled not figures.

Point 6. Figure 6. Please indicate what is illustrated in Figure (a) and (b).

Point 7. Figure 7. Is it important at Y-axis to give Percent with “.00”? Why do not give 100 instead of 100.00

Point 8. Figure 8. Please indicate what is illustrated in Figures (a) and (b).

Point 9. Figure 8. You should use dots instead of a comma across the figure.

Point 10. Section 5. Conclusions and Future Work. Maybe it will be better to name it “Conclusions and Future Study” or “. Conclusions and Future Investigation”

Point 11. Please consider the suggested research in your literature review:

Kalinichenko, V., Dolgikh, O., Dolgikh, L., & Pysmennyi, S. (2020). Choosing a camera for mine surveying of mining enterprise facilities using unmanned aerial vehicles. Mining of Mineral Deposits, 14(4), 31-39.

Jinqiang, W., Basnet, P., & Mahtab, S. (2021). Review of machine learning and deep learning application in mine microseismic event classification. Mining of Mineral Deposits, 15(1), 19-26.

Point 12. In general, I must admit that a very good study was performed.

Author Response

We thank the reviewer for the time and effort put into reviewing our paper. We also appreciate the constructive input and suggestions that the reviewer has provided. Please find attached our point-to-point response to the reviewer’s comments as well as descriptions of how the paper was revised.  At the end of this document, please find attached the revised manuscript with changes from the previous version marked in red.

Reviewer 2 Report

Dear Authors,

I have reviewed the paper entitled "Towards semantic photogrammetry: generating semantically rich point clouds from architectural close range photogrammetry". The manuscript deals with the integration of deep learning-based semantic image segmentation with 3D photogrammetric reconstruction and classification. In my opinion the paper is interesting but I have some issues that I would like the Authors to refer to.

Kind regards.

The Abstract could be better. For example, what is the scientific problem which the Authors noticed and solved in the article ? Moreover there should be information about the results or simple conclusions. It could interest the potential Reader into Your study.

Introduction Lines 17-32. The authors introduced 18 publications but to be honest I do not understand the meaning of every one of them. Maybe that is only what I am seeing this from my perspective. However I think that more consistent writing could be more interesting.

Line 33. Reference ? Proof ? Why point clouds and meshes became more common ?

Line 37. The same as above.

Line 46. Why does the problem of attribute annotation exist ?

Line 54. What is introduced in references No. 29 and No. 30?

Line 56. What do You mean by existing prior knowledge ?

In my opinion, the Introduction is unconsistent. The Authors should clear all of the problems and (what is the most important) Show the problem- solution chain with the simplest method possible. Many information that should be stated in the Introduction are in Chapter 2. Is it possible to merge them and better present the idea of the research ?

Chapter 3. The authors claimed that they dealt with "an integration of deep learning-based techniques”. What is the definition of an integration in Your case study ? It should be clearly stated in Chapter 3. In my opinion the results is OK, however, there are inconsistent reference formatting (Line 442).

Author Response

(The authors gave the same response as above.)

Round 2

Reviewer 1 Report

Dear authors,

I am more than satisfied with the corrections provided by you.

Congratulations to the authors.

Reviewer 2 Report

Dear Authors, 

Thank You for referring to all of my comments. I have no further questions.